# Correlation Study Between TV Viewing Variables and Cognitive Level, Depression Level, and Activities of Daily Living in Older Individuals Living Alone

**DOI:** 10.3390/healthcare13010016

**Published:** 2024-12-24

**Authors:** Sung Yeon Oh, Bum Sun Kwon, Yeon Gyo Nam

**Affiliations:** 1Department of Physical Therapy, Sun Moon University, Asan 31460, Republic of Korea; akk1263@naver.com; 2Department of Rehabilitation Medicine, Graduate School of Dongguk University, Seoul 04260, Republic of Korea; bumsunkwon@gmail.com; 3Digital Healthcare Institute, College of Health Sciences, Sun Moon University, Asan 31460, Republic of Korea

**Keywords:** aged, television, screen time, cognition, depression, activities of daily living

## Abstract

**Background/objectives:** Although there are studies on TV viewing and the health status of elderly, they do not present direct associations with specific variables. The aim of this study was to determine correlations between TV viewing variables and elderly health variables in older adults living alone. **Methods:** Data were collected from 50 elderly individuals with an average age of 82.12 ± 4.32 (Male 20%, Female 80%). Television viewing variables were collected via a TV set-top box, including the daily average viewing time, upper zapping threshold, lower zapping threshold, and average zapping per hour. The cognitive level was assessed using the Mini-Mental-State Examination, the depression level was assessed with the Geriatric Depression Scale, and activities of daily living were assessed using the Modified Barthel Index. Spearman correlation analysis was applied to the collected variables. **Results:** The results showed significant correlations between the depression level and both the daily average viewing time (r = 0.320) and upper zapping threshold (r = 0.308). The activities of daily living demonstrated significant correlations with the daily average viewing time (r = −0.313) and upper zapping threshold (r = −0.352). **Conclusions:** The TV viewing time and zapping are associated with depression and daily living activities, suggesting their potential as early diagnostic indicators for geriatric diseases in older adults living alone.

## 1. Introduction

The incidence of age-related diseases increases exponentially with advancing age, influenced by the accumulation of senescent cells that promote inflammation and impair the regenerative capacity. This process is associated with an annual rise in the age-related disease incidence of approximately 6–8% [1]. Globally, among individuals with two or more chronic conditions, prevalence rates reach approximately 37.2% of adults overall, affecting 44.4% of those aged 30 and older, 45.7% of those 40 and older, 47.2% of those 50 and older, and about 51% of those 60 and over [2]. Additionally, as the life expectancy increases, individuals are spending more time in advanced age, which can further elevate disease prevalence among older populations. In the United States, data from the Health and Retirement Study show that the proportion of older adults with at least one chronic condition rose from 86.9% in 1998 to 92.2% in 2008. During the same period, the percentage of older adults with four or more chronic diseases increased from 11.6% to 17.4% [3].

Geriatric diseases encompass a wide range of cognitive and physical conditions. Among them, cognitive decline, depression, and decreased activities of daily living (ADL) performances are particularly significant. Cognitive decline is closely related to neurodegenerative diseases such as Alzheimer’s disease, vascular dementia, and Parkinson’s disease, affecting memory, attention, and problem-solving abilities, thereby directly impacting the ability of older individuals to perform daily activities [4]. As of 2019, an estimated 51.6 million people globally, about 0.7% of the world’s population, are living with dementia. This represents more than a doubling of dementia cases since 1990, indicating a continuous rise in the global prevalence of dementia. Over the past 29 years, the prevalence, incidence, mortality, and disability-adjusted life years (DALYs) associated with dementia have steadily increased worldwide. The number of deaths due to dementia is projected to rise from approximately 2.4 million in 2019 to about 5.8 million in 2040 [5]. These statistics demonstrate that dementia has become a serious global health issue and predict a sharp increase in the burden on future medical and social care systems.

Depression and anxiety disorders are closely related to cognitive decline among older adults. These mental disorders can increase the risk of developing cognitive diseases such as dementia and, particularly, depression is often considered a prodromal symptom of dementia [6]. The prevalence of mental disorders, including depression and anxiety, among those aged 65 and over is about 20%, indicating that a significant portion of the elderly population is affected by mental health issues. The most common mental disorders among the elderly include depression, anxiety disorders, and in severe cases, psychotic symptoms such as hallucinations and delusions. Depression and anxiety disorders tend to occur at higher rates in the elderly compared to other age groups, with several studies reporting that approximately 6% to 12% of older adults experience anxiety disorders [7].

Furthermore, complications such as cognitive decline, depression, falls, and sarcopenia lead to decreased ADL function in the continuum of geriatric diseases [8]. ADL function measures the ability of older individuals to independently perform basic activities such as eating, bathing, dressing, and moving. A decline in ADL function makes independent living difficult, necessitating additional care and support. ADL disability and physical function decline are closely associated with depressive symptoms, and these outcomes are particularly pronounced among elderly individuals living alone in rural areas [9]. The prevalence of ADL disability among the community-dwelling elderly living alone increased by about 9% over five years in the 2000s, suggesting that the early resolution of physical function decline is important for the prevention and treatment of depression [10].

The early detection of geriatric diseases is crucial in slowing disease progression and preventing complications. Timely diagnosis and management can enhance the quality of life for older people, helping them maintain independence. However, for older adults living alone, the early diagnosis of geriatric diseases is particularly limited due to factors such as a lack of information, mobility issues, and communication difficulties [11]. Therefore, it is essential to develop more elderly-friendly methods for these individuals. Recently, innovative methods for early detection have been introduced. For example, various sensors that detect movement can continuously monitor activities within an elderly person’s living space, enabling the construction of models that detect deviations from normal activity patterns [12]. Additionally, by automatically learning appliance usage patterns, one can analyze the behavior of older adults through household power consumption and resident location data [13]. Although many such technologies have been developed, most require additional processes like installing new devices, making them difficult to access.

Therefore, this study aims to propose a new method by utilizing TV viewing variables that can be sufficiently collected in existing environments. Older people spend approximately 37% of their waking hours watching television, averaging 6.4 h per day [14]. This indicates that TV viewing occupies a significant portion of older adults’ lives. However, such viewing habits do not positively influence health. Prolonged TV watching is associated with an increased risk of dementia, a decline in cognitive functions such as language and memory, and various other negative outcomes for older individuals [15]. Moreover, limitations in physical function, mobility, and agility have been observed to increase with the TV viewing time, regardless of the physical activity level [16]. Thus, TV viewing and the health of older adults are closely linked, suggesting the possibility of analyzing the impact of TV viewing habits on their health status and predicting and responding to their physical and cognitive health conditions.

The high dependency of older adults on television suggests that analyzing viewing patterns can predict their health status. While the correlation between the TV viewing time and health has been sufficiently studied, it is just one of many variables that can be obtained from TV viewing. By identifying the times when TV viewing is frequent, one can determine the primary activity hours of the individual; the types of programs watched can also serve as indicators of their emotional state. Recent studies have explored new methods, such as predicting the depression levels of older people using their TV viewing time and program types [17], or developing health management algorithms for older adults using TV program types and motion sensors [18]. However, no research has directly examined the relationship between these TV viewing variables and the health of older adults. If a broader range of variables related to TV viewing could be analyzed to reveal and systematize the correlations with the cognitive and psychological states of older individuals, it might be possible to detect and diagnose diseases early using only TV viewing patterns.

Television significantly affects the lifestyle patterns of older individuals, and related data are being collected in real time through set-top boxes required for TV viewing. Although these data are collected from numerous households, no research has yet examined its correlation with health. In this study, we collect TV viewing variables from older adults living alone, such as the daily average viewing time, and zapping variables (the upper zapping threshold, lower zapping threshold, and average zapping per hour). This is basic information that can be obtained from commonly used set-top boxes at home [19]. The aim of this study was to determine the correlations between these television viewing variables and the elderly health variables, such as cognitive and depression levels, and ADL, in older individuals living alone.

## 2. Materials and Methods

### 2.1. Subjects

This study was conducted in collaboration with the Administrative Welfare Center in Paju City, Gyeonggi Province, and the real-time TV rating survey company ATAM Co., Ltd. (Seoul, Republic of Korea), a real-time TV ratings survey company. This study was designed as a cross-sectional observational study, following the STROBE (Strengthening the Reporting of Observational Studies in Epidemiology) guidelines to ensure robustness in design and reporting. Since there were no prior studies analyzing the correlation between TV viewing variables and health variables to determine the sample size for the experiment, we established the basis for the number of subjects in this study by averaging the sample sizes of four recent correlational studies on other topics.

The referenced studies are as follows. Ivanyshyn et al. involved 47 male adolescents aged 13–14 who participated in extreme motor activities, investigating the correlation between coordination abilities and mental properties [20]. Ruan included 50 college students, analyzing the relationship between linguistic logic and English writing proficiency [21]. Kim examined 51 stroke patients over 50 years old to assess the association between grip strength and pulmonary function [22]. Finally, Mozos et al. focused on 48 young adults, evaluating cardiovascular risk factors through pulse wave analysis [23]. By averaging the sample sizes of these studies, it was determined that a sample size of approximately 49 participants ensures consistency with existing correlational research.

In accordance with the determined sample size, we randomly selected 56 households of elderly individuals living alone who are managed by the Administrative Welfare Center of Paju. Participants aged 65 and older who consented to participate in this study were selected. Out of 56 individuals initially recruited, 6 were excluded due to missing data, resulting in 50 participants’ data being utilized for the final statistics.

The exclusion criteria for the subjects were as follows:(1)Patients experiencing memory disorders due to neurological abnormalities;(2)Patients with a history of Axis I psychiatric disorders, including intellectual disabilities, schizophrenia, alcohol dependence, and bipolar disorder;(3)Patients who had undergone cerebrovascular surgery;(4)Patients with a history of substance abuse within the past 5 years;(5)Patients with a history of alcohol addiction treatment within the past 5 years;(6)Patients with vision impairment who were unable to read normal text even with the use of glasses;(7)Patients with hearing impairments who found it difficult to understand conversations even with the use of hearing aids;(8)Other patients deemed unsuitable for participation in the clinical trial by the examiner.

Individuals with other conditions that could potentially impact cognitive or depression levels or ADL were excluded. Additionally, due to the use of manual measurement scales, individuals who had difficulty with communication or could not be assessed using these scales were excluded. Other individuals deemed unsuitable for this study by the investigator were also excluded.

This research received approval from the Institutional Review Board (IRB) of Sunmoon University (SM-202401-002-3). This study adhered to the ethical standards of the Declaration of Helsinki, ensuring compliance with current ethical and regulatory guidelines for studies involving human participants and safeguarding their rights, safety, and well-being throughout the research process.

### 2.2. Experimental Procedures

On the day the TV viewing data collection began, the researchers visited the homes of each elderly individual living alone and conducted all assessments for the participants.

General characteristics of the subjects, including age, gender, height, and weight, were also collected during assessments. Based on the collected data, we analyzed the correlations between participants’ TV viewing patterns and elderly health variables. This study was conducted according to the following procedure (Figure 1).

### 2.3. TV Viewing Variables

In this study, the correlations between TV viewing variables and elderly health variables were identified.

To collect TV viewing variables, set-top boxes installed in the homes of older individuals living alone, managed by the Administrative Welfare Center and community centers in Paju City, Republic of Korea, were utilized. These set-top boxes, connected to the participants’ TVs for care and welfare purposes, recorded the viewing information. The TV viewing variables used in the analysis of this paper were basic variables that can be obtained from any set-top box. The collected viewing data processed for statistical analysis in this study spanned 3 months, from 1 October 2023 to 31 December 2023.

The TV viewing variables analyzed in this study included daily average viewing time (DAV), upper zapping threshold (UZT), lower zapping threshold (LZT), and average zapping per hour (AZH). DAV was measured as the total viewing minutes, while zapping was defined as the act of switching channels using a remote control, either to avoid unwanted content or to explore other programs [24]. The number of zapping instances was averaged at hourly intervals, and all data classification methods were conducted in collaboration with ATAM Co., Ltd. AZH was calculated as the daily average number of zapping instances. UZT referred to the instances when the current data exceeded the maximum value recorded over the past four weeks, while LZT captured instances when the current data fell below the minimum value within the same timeframe. These comprehensive variables provided detailed insights into TV viewing patterns among participants.

### 2.4. Elderly Health Variables and Instrumentation

The elderly health variables collected in this study are cognitive level (CL), depression level (DL), and activities of daily living (ADL).

To assess these variables, three validated assessment scales were employed: the Mini-Mental-State Examination (MMSE) for CL assessment, the Geriatric Depression Scale (GDS) for evaluating the DL, and the Modified Barthel Index (MBI) for ADL. All assessment scales were administered manually. The researcher visited the participants’ homes to provide explanations regarding the scales, and participants completed the assessment scales prepared in a paper-and-pencil format.

#### 2.4.1. MMSE

Developed by Folstein and McHugh in 1975, the MMSE is a cognitive assessment scale that is easier to administer than other cognitive scales and can be completed in 5–10 min. It has the advantage of minimal practice effects, allowing for repeated measurements throughout a disease to observe changes over time [25]. The scoring includes 10 points for temporal and spatial orientation (5 points each), 3 points for memory registration, 3 points for recall, 5 points for attention and calculation, 7 points for language function, and 2 points for comprehension and judgment, providing a total of 30 points. A score of 24 or above is considered “no cognitive impairment”, 20–23 is “mild cognitive impairment”, and 19 or below is “severe cognitive impairment”. For illiterate individuals, 1 point for temporal orientation, 2 points for attention and calculation, and 1 point for language function are added, ensuring that the total does not exceed the maximum score for each section [26]. The MMSE is widely used across various fields for its high reliability and validity as a simple cognitive function screening tool [27]. The Korean version of the MMSE (MMSE-K) was used in this study to accommodate the participants’ language. The MMSE-K has been shown to be reliable and valid [28], with demonstrated usage across various studies [29].

#### 2.4.2. GDS

The GDS is a screening tool used to evaluate the severity of depressive symptoms in individuals aged 65 and above [30]. The original scale includes 30 items, while shorter versions contain 20, 15, 12, 10, 5, and 4 items. The scale is useful in distinguishing not only depressive disorders in older adults, but also cognitive impairments, including dementia and physical illnesses that do not manifest as depressive symptoms. In this study, the scale consisted of 30 items, each scoring 1 point, allowing for a maximum score of 30. According to the score criteria, 0–9 points indicate a normal mood state, 10–19 points indicate mild depressive symptoms, and 20–30 points indicate severe depressive symptoms [31]. The scale is a useful screening and monitoring tool in primary care settings due to its high reliability, validity, sensitivity, and specificity [32]. The Korean version of the GDS (KGDS) was used in this study to accommodate the participants’ language. The KGDS has been shown to be reliable and valid [33], with demonstrated usage across various studies [34].

#### 2.4.3. MBI

The MBI is a widely used scale for assessing ADL performance, and it is reported to have high sensitivity, simplicity, ease of scoring, and high reliability and validity compared to other tools [35]. It consists of 11 items concerning the following ADLs: personal hygiene, bathing, eating, toilet use, stair climbing, dressing, bowel control, bladder control, ambulation, wheelchair use, and transfers (chair/bed). The wheelchair item is measured only if the individual cannot walk; thus, the actual number of items measured is 10. Each item is scored on a 5–15-point scale, ranging from complete dependence to complete independence, where performing all items completely independently scores 100 points [36]. The Korean version of the MBI (KMBI) was used in this study to accommodate the participants’ language. The KMBI has been shown to be reliable and valid [37], with demonstrated usage across various studies [38].

### 2.5. Statistical Analysis

In this study, descriptive statistics were utilized to calculate general characteristics, including the mean and standard deviation of each variable. The Kolmogorov–Smirnov test was performed to examine the normality distribution of the collected data. The correlations between variables were analyzed using Spearman’s rank correlation coefficient (rs), which measures the strength of the linear relationship between two variables, where +1 indicates a positive linear relationship, −1 indicates a negative linear relationship, and 0 indicates no linear relationship. Values above 0.9 are considered very high, between 0.7 and 0.9 high, between 0.5 and 0.7 moderate, between 0.3 and 0.5 low, and below 0.3 indicates little, if any relationship, applicable to both positive and negative cases [39].

Since correlation variables only measure the association between two variables, it cannot be determined whether TV viewing variables actually impact elderly health variables. Therefore, an independent *t*-test was conducted by dividing the variables with significant correlations into two groups. The grouping was performed as follows: each variable’s mean was calculated, and participants were categorized into groups higher and lower than the mean.

All statistical analyses were conducted using IBM SPSS 26.0 statistical software. The degrees of freedom for all correlation analyses were calculated as n − 2, where n is the sample size. Statistical significance levels were set at *p* < 0.05.

## 3. Results

### 3.1. General Characteristics of Subjects

A total of 50 individuals aged 65 and above participated in the study. The mean age of participants was 82.12 ± 4.32. The average height was 156.94 ± 6.59 cm, and the average weight was 56.76 ± 9.99 kg. The gender included 10 males (10%) and 40 females (80%). The mean CL was 23.72 ± 4.24. The mean DL was 12.60 ± 6.94. The mean ADL was 90.02 ± 21.28. The DAV was 639.83 ± 297.21. The UZT was 34.82 ± 27.79. The LZT was 0.62 ± 1.39. Finally, the AZH was 3.09 ± 1.65 [Table 1].

### 3.2. Analysis of Correlations Between TV Viewing Data and Elderly Health Variables

According to Spearman’s correlation results, no significant correlations were found between CL and the other variables. The correlation coefficient between DL and DAV was r(48) = 0.320, indicating a low positive correlation, which was statistically significant (*p* < 0.05). Similarly, the correlation coefficient between DL and the UZT was r(48) = 0.308, indicating a low positive correlation, and it was statistically significant (*p* < 0.05). No significant correlation was found between DL and the other variables (*p* > 0.05). The correlation coefficient between ADL and DAV was r(48) = −0.313, indicating a low negative correlation, which was statistically significant (*p* < 0.05). Likewise, the correlation coefficient between DL and the UZT was r(48) = −0.352, indicating a low negative correlation, and it was statistically significant (*p* < 0.05). No significant correlations were found between ADL and the other variables (*p* > 0.05). The correlation coefficient between the DAV and the UZT was r(48) = 0.812, indicating a high positive correlation, which was statistically very significant (*p* < 0.01). The correlation coefficient between the AVH and the UZT was r(48) = 0.495, indicating a moderate positive correlation, which was also statistically very significant (*p* < 0.01) (Table 2).

### 3.3. Independent Test Based on DAV Group Classification

A comparison between the HDAV and LDAV groups was conducted using an independent *t*-test. Significant differences were found in the UZT, AZH, and DL (*p* < 0.05), while no significant differences were observed for the remaining variables (*p* > 0.05) (Table 3).

### 3.4. Independent Test Based on UZT Group Classification

A comparison between the HUZT and LUZT groups was conducted using an independent *t*-test. Significant differences were found in the DAV, AZH, and DL (*p* < 0.05), while no significant differences were observed for the remaining variables (*p* > 0.05) (Table 4).

## 4. Discussion

In this study, we analyzed the correlation between TV viewing patterns and elderly health variables of the elderly aged 65 and over who live alone. Spearman correlation analysis revealed significant correlations between the DAV, AZT and DL, and ADL. In the additional independent *t*-test conducted, the comparison between the HDAV and LDAV showed significant differences in the UZT, AZH, and DL (*p* < 0.05). Similarly, in the comparison between the HUZT and LUZT, significant differences were observed in the DAV, AZH, and DL (*p* < 0.05).

As the starting point of media technology, TV has brought about significant changes in human culture since its introduction. These changes have influenced not only the realm of entertainment, but also human daily life and health. As TV viewing has become an essential part of everyday life, the necessity for research on issues related to TV viewing and one’s health status has emerged. Previous studies have illuminated the correlation between TV viewing and one’s health status from various angles, including cognitive and physical aspects. For instance, according to the study by Neto et al., higher TV viewing times are associated with lower physical activity, leading to a higher body mass index, waist circumference, body fat percentage, glucose, triglycerides, and HDL cholesterol levels, thus increasing the risk of obesity and various heart diseases [40]. Moreover, TV has a close association with mental health, as the study by McAnally et al. revealed that excessive TV viewing during childhood and adolescence increases the risk of developing anxiety disorders in older people [41]. These findings suggest that TV viewing, beyond a mere leisure activity, can significantly impact an individual’s physical and mental health.

There is abundant research on the outcomes of TV viewing, such as decreased physical activity and increased psychological instability. However, few studies have been conducted on predicting users’ health through easily collectible TV viewing variables. Some research studies have solely analyzed the TV viewing time [17], while others have utilized applications [42] or employed wearable cameras simultaneously to gather information beyond TV viewing [43]. However, this study distinguishes itself from other studies by exclusively utilizing variables obtained from TV viewing to elucidate their relevance to health information. Assuming that zapping data, including the total viewing time, viewing time slots, excessive zapping, and insufficient zapping, would vary according to the user’s cognitive, depression, and ADL performance levels, we analyzed the correlations among these variables and derived several significant findings. Among these, the linkage between the degree of depression, the level of ADL performance, and zapping data is particularly meaningful.

This study’s results also include findings consistent with previous research. The correlation coefficient between DL and DAV was 0.320, indicating a low positive correlation and statistical significance. This suggests that longer TV viewing times are associated with higher levels of depression among users. Moreover, the independent *t*-test comparing HDAV and LDAV showed a significant difference in DL. This implies that depression levels can change based on regular TV viewing times. Santos et al. investigated the associations among TV viewing, physical activity, and depressive symptoms in European adults, finding that participants who watched TV for more than 2 h a day had higher depressive symptom scores, which could be mitigated through physical activity [44]. Tolba et al.’s study focused on “binge-watching” and analyzed its associations with depression and loneliness. This study found that binge-watching is associated with both depression and loneliness, particularly noting that some viewing motives were related to depression and loneliness, suggesting that binge-watching can negatively impact mental health [45]. Additionally, Yu et al. analyzed the associations between computer/mobile device use, TV viewing, and depression, finding that longer TV viewing times were associated with a higher likelihood of depressive symptoms [46]. These results are consistent with the findings of this study. Furthermore, when examining TV viewing variables in the future, there is an implication that we can predict an increase in the level of depression when a pattern of an increasing TV viewing time is observed.

Additionally, the statistical analysis results of this study showed a low negative correlation coefficient of −0.313 between ADL and DAV, which was statistically significant. This indicates that longer TV viewing times correspond to a lower ADL. According to Fingerman et al., physical function limitations in older adults lead to an increase in the TV viewing time. This is linked not only to decreased social, physical, and productive activities, but also suggests the potential for further disabilities or a loss of autonomy [47]. García-Esquinas et al. investigated the expected associations between the TV viewing time and physical function, mobility, agility, and frailty indicators among older adults in a cohort study, revealing that longer TV viewing times in older adults, regardless of physical activity, are associated with a higher likelihood of physical function limitations [16]. Furthermore, Lin et al. examined the associations of physical activity, muscle-strengthening activities, and TV viewing, finding that excessive TV viewing negatively affects the physical activity of older adults, particularly showing low adherence to muscle-strengthening activities. This suggests a potential association between TV viewing and physical function decline in older adults [48]. Again, these research findings are consistent with the results of this study. This also suggests, similar to the previous content, that through the analysis of TV viewing patterns, we can predict a decrease in ADL when the TV viewing time increases.

A noteworthy point in this study was the relationship between the zapping and health variables. The statistical analysis revealed a low positive correlation coefficient of 0.308 between the DL and UZT, which was statistically significant. Similarly, the correlation coefficient between ADL and the UZT was −0.352, indicating a low, statistically significant negative correlation. This suggests that excessive zapping occurs among users with higher levels of depression or reduced physical function. Such findings could be attributed to the anxiety accompanying depression. Generally, higher levels of depression are associated with higher anxiety scores, and factors such as a loss of pleasure, dependency, low self-confidence, isolation, medical conditions, poor financial status, memory problems, suffering, identity issues, physical limitations, fear of death, and loneliness contribute to increased depression and anxiety among older adults [49]. To reduce this anxiety, behavioral symptoms such as leg shaking and pacing may occur [50]. These behavioral symptoms are physical reactions commonly observed in states of anxiety and can include zapping as an expression of the physical tension experienced by individuals feeling anxious. Moreover, the independent *t*-test comparing the HUZT and LUZT showed a significant difference in DL, which adds more reliability to the above analysis.

Additionally, a decrease in physical activity due to physical function decline can increase anxiety levels. Khodabakhshi-Koolaee et al. investigated loneliness and anxiety according to physical activity among bedridden older individuals, finding higher loneliness and anxiety scores in the less active group [51]. These research findings further support the results of this study. This result was not found in previous studies. When analyzing TV viewing patterns, we need to recognize that if the number of zapping instances is high or increasing, the level of depression may be higher, or the ADL may be lower. Such a pattern proves that TV viewing variables can be effective in predicting the health of older adults.

This study has several limitations. First, it considered the time the TV was used rather than the actual TV viewing time. This is because only data on the TV’s ON and OFF status were collected, which did not account for situations where the viewer was engaged in other activities while the TV was on or when anomalies occurred. Secondly, the sample size is relatively small. A small sample size may limit the generalizability of the study findings. Further research with larger and more diverse populations is necessary to validate the use of TV viewing patterns. If confirmed, this approach could offer significant clinical benefits, particularly for the remote monitoring of elderly individuals living alone.

## 5. Conclusions

This study was aimed to collect data on detailed variables related to TV viewing and analyze their correlations with elderly health variables. The statistically significant results were as follows: a low positive correlation was found between the ADL and DAV and the UZT, while a low negative correlation was found between the ADL and DAV and UZT. Additionally, when the DAV and UZT were divided into groups for comparison, a significant difference was observed with the DL. These findings suggest that geriatric diseases can be predicted through the collection of TV viewing variables. This is expected to be particularly effective for older individuals living alone with limited healthcare access. Future research needs to collect a larger sample size to conduct regression analysis, clarifying whether cognitive and psychological factors drive the increased TV viewing time or whether extended TV viewing impacts cognitive and psychological outcomes.

## Figures and Tables

**Figure 1 healthcare-13-00016-f001:**
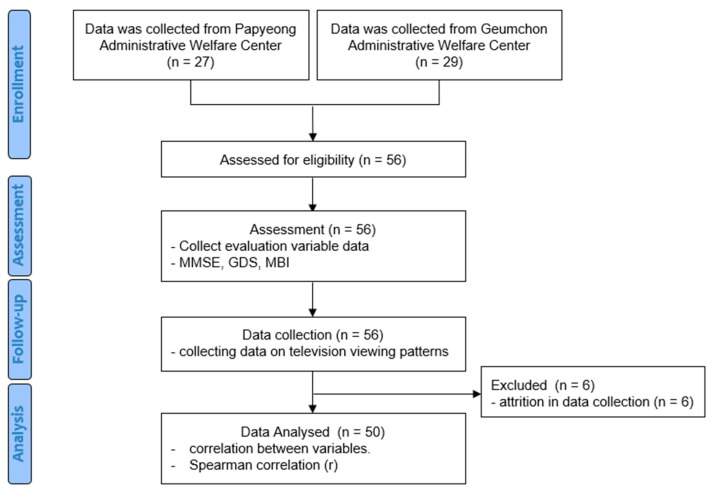
Flowchart of the experimental procedure.

**Table 1 healthcare-13-00016-t001:** Participants’ general characteristics.

	Participants (n = 50)
Age (years)	82.12 ± 4.32
Height (cm)	156.94 ± 6.59
Weight (kg)	56.76 ± 9.99
CL (MMSE score: 0–30)	23.72 ± 4.24
DL (GDS score: 0–30)	12.60 ± 6.94
ADL (MBI score: 0–100)	90.02 ± 21.28
DAV (min)	639.83 ± 297.21
UZT (number of days)	34.82 ± 27.79
LZT (number of days)	0.62 ± 1.39
AZH	3.09 ± 1.65

Values are mean ± SD. DAV, daily average viewing time; UZT, upper zapping threshold; LZT, lower zapping threshold; AZH, average zapping per hour; CL, cognitive level; DL, depression level; ADL, activities of daily living.

**Table 2 healthcare-13-00016-t002:** Correlations between TV viewing data and cognitive function, depression, and ADL performance.

		DAV	UZT	LZT	AZH	CL	DL	ADL
DAV	r		0.812 **	0.150	0.495 **	0.264	0.320 *	−0.313 *
	*p*		0.000	0.374	0.000	0.128	0.032	0.013
UZT	r	0.812 **		0.046	0.590 **	0.145	0.308 *	−0.352 *
	*p*	0.000		0.850	0.000	0.472	0.038	0.006
LZT	r	0.150	0.046		0.324	0.033	0.098	−0.127
	*p*	0.374	0.850		0.399	0.718	0.547	0.380
AZH	r	0.495 **	0.590 **	0.324		0.087	0.218	0.088
	*p*	0.000	0.000	0.399		0.508	0.169	0.708
CL	r	0.264	0.145	0.033	0.087		−0.249	−0.091
	*p*	0.128	0.472	0.718	0.508		0.110	0.755
DL	r	0.320 *	0.308 *	0.098	0.218	−0.249		−0.153
	*p*	0.032	0.038	0.547	0.169	0.110		0.220
ADL	r	−0.313 *	−0.352 *	−0.127	0.088	−0.091	−0.153	
	*p*	0.013	0.006	0.380	0.708	0.755	0.220	

* *p* < 0.05, ** *p* < 0.01. DAV, daily average viewing time; UZT, upper zapping threshold; LZT, lower zapping threshold; AZH, average zapping per hour; CL, cognitive level; DL, depression level; ADL, activities of daily living.

**Table 3 healthcare-13-00016-t003:** Independent Test Based on DAV Group Classification.

	UZT	LZT	AZH	CL	DL	ADL
HDAV (n = 25)	55.16 ± 24.32	0.64 ± 1.08	3.74 ± 1.85	24.40 ± 3.74	14.80 ± 7.23	84.84 ± 27.74
LDAV (n = 25)	16.00 ± 15.52	0.60 ± 1.68	2.38 ± 1.08	22.80 ± 4.73	10.60 ± 5.82	94.96 ± 10.41
t	−6.788	−0.100	−3.190	−1.327	−2.261	1.708
*p*	0.00 **	0.92	0.00 **	0.19	0.03 *	0.09

* *p* < 0.05, ** *p* < 0.01. HDAV, high group of daily average viewing time; LDAV, low group of daily average viewing time; UZT, upper zapping threshold; LZT, lower zapping threshold; AZH, average zapping per hour; CL, cognitive level; DL, depression level; ADL, activities of daily living.

**Table 4 healthcare-13-00016-t004:** Independent Test Based on UZT Group Classification.

	DAV	LZT	AZH	CL	DL	ADL
HUZT (n = 22)	861.20 ± 149.17	0.45 ± 0.86	4.02 ± 1.79	23.64 ± 4.07	15.41 ± 7.46	86.77 ± 23.53
LUZT (n = 28)	476.11 ± 282.36	0.75 ± 1.71	2.30 ± 1.04	23.57 ± 4.54	10.57 ± 5.56	92.36 ± 19.57
t	−5.785	0.738	−4.256	−0.053	−2.629	0.916
*p*	0.00 **	0.46	0.00 **	0.96	0.01 *	0.36

* *p* < 0.05, ** *p* < 0.01. HUZT, high group of upper zapping threshold; LUZT, low group of upper zapping threshold; DAV, daily average viewing time; LZT, lower zapping threshold; AZH, average zapping per hour; CL, cognitive level; DL, depression level; ADL, activities of daily living.

## Data Availability

The data presented in this study are available on request from the corresponding author due to privacy restrictions. Please contact akk1263@naver.com for access.

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
