# Peer review of "Correlation Study Between TV Viewing Variables and Cognitive Level, Depression Level, and Activities of Daily Living in Older Individuals Living Alone"

_healthcare, 2024, doi:10.3390/healthcare13010016_

Round 1

Reviewer 1 Report (Previous Reviewer 2)

Comments and Suggestions for Authors Abstract: what is  ADL levels? line 452:  At 4 ??? use AM where is necessary  

Author Response

Thank you for carefully reviewing our manuscript. We hope this revision letter meets your expectations.

Reviewer 2 Report (Previous Reviewer 1)

Comments and Suggestions for Authors

This manuscript entitled “Correlation Study Between TV Viewing Patterns and Cognitive Function, Depression, and Activities of Daily Living in Older Individuals Living Alone” was primarily aimed to the potential of TV viewing patterns for early detection of geriatric diseases. Authors bring an interesting study, but there are still some problems that cannot up this article to a publishing level. Suggestions are listed in the specific comments below.

Specific comments:

1.     In the Abstract part, line 20-21, “…indicating that individuals with lower ADL levels tend to spend more time watching TV…”. Please give the official explanation of the abbreviation “ADL” here

2.     In the first paragraph of the Introduction part, providing this data here is a great way to draw attention to chronic diseases in older adults. But authors are recommended to provide which country does this data come from. Or is it worldwide?

3.     In the Introduction part, line 184-185, “The types of programs watched and the frequency of channel changes can also serve as indicators of emotional state.” Please cite the relevant paper to support this statement.

4.     In the Materials and Methods part, Statistical Analysis, please add the information about the significance level.

5.     In the Discussion part, line 497-498, “Previous studies have illuminated the correlation between TV viewing and health status from various angles, including cognitive and physical aspects.”. Please cite the relevant papers to support this statement. Some recently studies could be added in the discussion, such as: A Comparison of Physical Activity and Sedentary Lifestyle of University Employees through ActiGraph and IPAQ-LF’, Physical Activity and Health, 6(1), p. 5–15. Available at: https://doi.org/10.5334/paah.163.

6.     In the Conclusion part, line 703, “This study aimed to collect data on detailed variables related to…” Please replace “aimed” with “was aimed”.

Comments on the Quality of English Language

English writing is fine. 

Author Response

Thank you for carefully reviewing our manuscript. We hope this revision letter meets your expectations.

Reviewer 3 Report (New Reviewer)

Comments and Suggestions for Authors

Dear Authors,

After reviewing your manuscript, it is observed that you present an innovative research regarding the methodology to determine TV time. However, it is necessary to improve some key methodological aspects in the report of the study, as well as the research objective, which in its current formulation does not fully respond to the knowledge gap that is addressed. Below, I detail comments that I hope will contribute to improving the quality of your manuscript:

Abstract:

The objective of the study is not completely clear. It is recommended that you mention it explicitly using a formulation such as "The aim of this study was...".

It would be useful to briefly include background information that reflects the problem and the "gap" that the study addresses.

Basic information should be provided on the methodology used, such as the study design, characteristics of the population (sample size, average age, and percentage by gender), and the measurement instruments used.

It is important that the study variable be mentioned instead of focusing solely on the measurement instruments. This last comment, please consider for the entire manuscript.

Introduction

The evidence presented by the authors suggests the impact of TV viewing on health variables. However, the objective proposed in this study, if achieved, would not fully answer the research question posed, since a correlation analysis is chosen that only provides evidence on bilateral relationships. In the case of finding a correlation between TV time and depression, it would not be clear whether depression drives participants to watch more TV or whether TV time influences the appearance of depression symptoms. Therefore, a regression analysis, with TV time as a predictor variable, would be more appropriate to clarify this relationship. Given the small sample size, it would also be convenient to consider t-tests, using TV time as a grouping variable. However, by increasing the sample to three or more groups, a problem of insufficient size could arise in some groups, which would make the analysis difficult to feasibility.

Materials and methods

It is essential to declare the design of the study, as well as to indicate whether any methodological guidelines were followed to ensure a robust structure, such as STROBE or another similar one. Since we worked with a company, it is necessary to clarify and argue that there was no conflict of interest, mentioning it in the corresponding section.

For the calculation of the sample size, the type of analysis planned must be considered as a relevant factor. However, this may vary depending on the statistical criteria adopted or the number and type of variables considered. Therefore, it is recommended to incorporate the statistical calculation of the sample size a priori.

Line 219: It would also be appropriate to mention that current ethical and regulatory guidelines for studies in humans and social sciences were followed, such as those of the APA or the Declaration of Helsinki.

Line 223: This does not constitute an inclusion criterion; it is an implicit aspect of the research.

Since numerous exclusion criteria are listed, it would be convenient to justify their inclusion in the study beforehand.

Data collection: The information on TV data collection is clear and detailed; it could be considered abbreviating it to include only the most relevant. On the other hand, the collection procedures for the other variables have not been described. It is important to specify how the questionnaires were applied: were they in paper and pencil format, online, or digital format? Were participants guided during the response? Did they respond from home?

Variables and instruments: The variable itself should be mentioned as the title, not the instrument used. In addition, for all instruments used, it is important to indicate whether they are validated in the language and population of the study. Their psychometric indicators, if they exist, should also be declared in previous studies in a similar population.

Line 344: Information on the statistical software used is less relevant and should be included at the end of the statistical analysis paragraph along with the level of significance used.

Statistical analysis: Please explain why Kruskal-Wallis analysis was used, as it does not seem to meet the study objective. Furthermore, this analysis is not suitable for groups with only one case. To obtain valid and stable results, each group should contain more than one case, preferably five or more, to ensure the statistical validity of the range distributions compared.

Results

Table 1: In the methodology section, it was not reported that sociodemographic information would be collected. It is necessary to mention how this information was obtained in that section of the manuscript.

Lines 405-406: This information was already presented in the statistical analysis section, so it should be removed from this part.

Why is the positive correlation between daily physical activity and time in front of the TV not mentioned? According to the results in the table, the more time in front of the TV, the more time of physical activity in daily life. This is a finding that can be complex to interpret and that the authors will have to address. It is possible that, by having the TV on, the participants were not necessarily sitting watching it; if they live alone, it is likely that they used it as company while doing other activities. However, in the case of zapping time, this could indicate that they were indeed in front of the TV.

When reporting the results, not only the significance value or p value should be indicated. It is also important to report the value of the test and the degrees of freedom. In addition, it is only reported that there are significant differences between the groups, but these differences are not necessarily present in all groups, but possibly only in some. To clarify this, it is also necessary to report the post hoc analyses.

The information in tables 4 and 5 does not respond to the objective of the study, so they should be eliminated.

Author Response

Thank you for carefully reviewing our manuscript. We hope this revision letter meets your expectations.

Round 2

Reviewer 2 Report (Previous Reviewer 1)

Comments and Suggestions for Authors

All my questions have been well addressed, now I recommend to accept. 

Comments on the Quality of English Language

The English could be improved to more clearly express the research.

Author Response

Thank you for making a meaningful decision based on my response.

This manuscript is a resubmission of an earlier submission. The following is a list of the peer review reports and author responses from that submission.

Round 1

Reviewer 1 Report

Comments and Suggestions for Authors

1. It would be helpful to elaborate on why television viewing patterns were specifically chosen as a focus for early diagnosis and to discuss how previous research has linked these patterns to health issues.

2. Please provide the reasons for choosing these variables.

3. Please clarify the basis for the sample size selection. Has the potential impact of gender and age range on the results been considered?

4. When comparing your results with other studies, it would be beneficial to discuss in more detail the similarities or differences in research methods or outcomes.

5. The study suggests using related indicators as predictive factors, but should the generalizability of these findings be cautiously considered given the sample size? and some recently studies shall be considered, such as: The Relationship between Functional Fitness and Ability to Ride a Bicycle among Community-Dwelling Older Adult in Japan’, Physical Activity and Health, 5(1), p. 45–54.

6. The data collection for TV viewing patterns and the completion of the assessment scales were not synchronizedhow can the validity of using these data for correlation analysis be justified?

Comments on the Quality of English Language

Minor editing of English language required.

Author Response

Thank you for your revision. I have incorporated your comments into the revision letter, which I have attached as a file.

Reviewer 2 Report

Comments and Suggestions for Authors -Abstract: "Depression Scale (KGDS) and daily average viewing time and excessive channel switching
were 0.320 and 0.308, respectively. The Korean version of the Modified Barthel Index (KMBI)
showed correlations of -0.313 with daily average viewing time and -0.352 with excessive channel 1. Introduction: switching.', why have you put the results here? what are the meaning of them? -"The Korean version of the Mini-Mental State Examination (MMSE-K) groups showed significant differences at 3, 4, and 5 PM.", what are 3,4 and 5 here?what are meaningful results of these findings? -"The KGDS  and KMBI groups showed significant differences at 8 PM and midnight and 1 AM and 2 AM, respectively.", please present clear and understandable results here that when one a person reads first and can understand it. Materials and Methods - is 50 sample size enough? what is your sampling strategy? Have you found them randomly? how?   Results -Line 270: Explain groups ' MMSE-K groups were found at 3, 4, and 5`, what are day? Line 238: where have you used  Korean version of the Modified Barthel Inde? and why?   What will be the contribution of your findings for policy makers, healthcare staffs or others?    

Comments on the Quality of English Language

is ok

Author Response

(The authors gave the same response as above.)

Reviewer 3 Report

Comments and Suggestions for Authors

Thank you for the opportunity to read and revise this manuscript. The research addresses the correlations between TV viewing patterns and several indicators of geriatric diseases. Although the manuscript explores an interesting topic, I have some concerns about this work. Below are my comments. The authors provided only a short description of geriatric diseases (lines 41-51). I suggest a widely improvement of this point, given the topic covered by this manuscript. Then, the authors included some information about innovative methods, which can help to early detection of geriatric disease. Given the aim of the study, I wonder why the authors introduced this information. The authors stated that the study directly examined the relationship between TV viewing variables and the health of older adults. However, they did not justify with well-defined literature why. Which is the rationale behind the study? No theoretically-grounded models have been included to explain the study. The study is lacking in research hypotheses. The novelty of this research should also be better disclosed and what this study adds to the current literature. As for the materials and methods, in the subject section the authors did not include relevant information about the sample. I understand that all this information is provided in Table 1. However, this paragraph should include information about the main features of the sample. More information should be provided about the rationale behind the chosen methodology. A deeper improvement is necessary in this section to enhance the reproducibility of your study. In the discussion, the authors should thoroughly examine the implications of their findings, especially in relation to existing literature. This would greatly enhance the depth of this section.

Comments on the Quality of English Language

The article requires moderate English revision.

Author Response

(The authors gave the same response as above.)
